# Characterization and Water Content Estimation Method of Living Plant Leaves Using Terahertz Waves

**Adnan Zahid** [1,*] , **Hasan T. Abbas** [1] , **Muhammad A. Imran** [1] , **Khalid A. Qaraqe** [2] ,
**Akram Alomainy** [3] , **David R. S. Cumming** [1] and **Qammer H. Abbasi** [1,*]

1    James Watt School of Engineering, University of Glasgow, Glasgow G12 8QQ, UK
2    Department of Electrical and Computer Engineering, Texas A&M University, Doha 23874, Qatar
3    School of Electronic Engineering and Computer Science, Queen Mary University of London,
     London E1 4NS, UK
*    Correspondence: a.zahid.1@research.gla.ac.uk (A.Z.); Qammer.Abbasi@glasgow.ac.uk (Q.H.A.)

**Abstract:** An increasing global aridification due to climate change has made the health monitoring of vegetation indispensable to maintaining the food supply chain. Cost-effective and smart irrigation systems are required not only to ensure the efficient distribution of water, but also to track the moisture of plant leaves, which is an important marker of the overall health of the plant. This paper presents a novel electromagnetic method to monitor the water content (WC) and characterisation in plant leaves using the absorption spectra of water molecules in the terahertz (THz) frequency for four consecutive days. We extracted the material properties of leaves of eight types of pot herbs from the scattering parameters, measured using a material characterisation kit in the frequency range of 0.75 to 1.1 THz. From the computed permittivity, it is deduced that the leaf specimens increasingly become transparent to the THz waves as they dry out with the passage of days. Moreover, the loss in weight and thickness of leaves were observed due to the natural evaporation of leaf moisture cells and change occurred in the morphology of fresh and water-stressed leaves. It is also illustrated that loss observed in WC on day 1 was in the range of 5% to 22%, and increased from 83.12% to 99.33% on day 4. Furthermore, we observed an exponential decaying trend in the peaks of the real part of the permittivity from day 1 to 4, which was reminiscent of the trend observed in the weight of all leaves. Thus, results in paper demonstrated that timely detection of water stress in leaves can help to take proactive action in relation to plants health monitoring, and for precision agriculture applications, which is of high importance to improve the overall productivity.

**Keywords:** vegetation health monitoring; leaf water content; terahertz; sensing; plants health

## 1. Introduction

Over the past decade, terahertz (THz) technology has seen an increased amount of interest in the scientific community chiefly due to its non-ionising and less pervasive radiation properties [1]. There has been significant progress in tapping the so-called THz gap 0.3 THz to 3 THz of the electromagnetic spectrum. The THz technology has found extensive use in applications such as the imaging of concealed items [2], material characterisation [1], diagnostic applications including treatment of skin and dental care [3,4], effective and quality control of food [5], and telecommunication [1,6,7]. Furthermore, a distinguishing feature of the THz waves is that the water molecules exhibit a strong absorption spectrum in the pertinent frequency range, leading to novel bio-sensing applications.

Despite these substantial contributions, the utility of the THz technology in the environmental control/monitoring systems has not been explored in depth, especially for vegetation monitoring [8,9]. Unlike the microwave-based remote sensing techniques, the THz technology can provide detailed

insight into the health of a plant specimen in terms of the water content (WC) in the leaves [10]. Plant leaves comprise of a composite biological structure of tissues, distinct bio-molecules like cellulose and synthesis compounds including proteins, carbohydrates and many other molecular weight compounds, as illustrated in Figure 1. On an individual basis, they vastly differ in terms of material properties such as relative permittivity [11]. Furthermore, water is not only an essential component but an important nutrient to the process of photosynthesis, and transpiration in the overall process of growth [12]. Due to the high sensitivity and strong penetration feature of THz, it has a strong potential to disseminate through plants leaves at cellular level as shown in Figure 1 and can yield significant information of WC in leaves. Hence, it is significant to highlight the frequency dependence of the permittivity of leaves. Designing a smart and plant-specific irrigation system that monitors the leaf WC in a non-invasive manner is, therefore, critical in the current circumstances governed by global climate change that demand water conservation. Over the years, significant contributions have been made [9–13], that address estimating the leaf's WC. There are techniques that offer high reliability, yet they are unsuitable for long-term studies of the same plant leaves because of validity of measurements cannot be guaranteed [11,14–17].

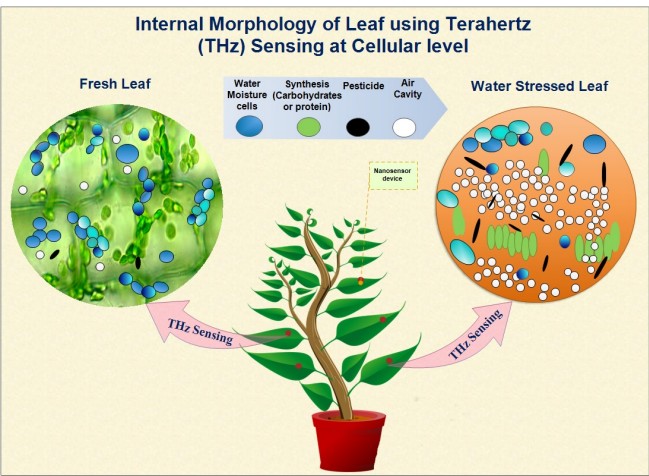

**Figure 1.** Internal morphology of fresh and water stressed leaf using THz sensing.

Thermogravimetric analysis has been used for the quantification of WC in plants leaves. However, due to its destructive nature and problems owing to its harmfulness to specimens have markedly reduced the usefulness, and thus is not suitable for estimation of WC in plants leaves. On the other hand, some non-destructive methods have previously been used to determine the water status in plant leaves, which include thermal and hyper-spectral imaging [18,19], infrared [20] and magnetic resonance imaging (MRI) [21]. However, these techniques are limited by the resolution and thus, cannot provide any cellular-scale information about the plants [19]. In addition, these technique do not consider any environmental influence due to its low photon energy [14,19]. Lately, there has been a growing trend in the field of plant physiology and characterisation of liquid to use THz time-domain spectroscopy (TDS) [11,22], which is considered a non-invasive technique, and has been deployed in the field of plant physiology to detect anomalies proactively. Moreover, it has an enormous potential to measure the leaf water status under certain conditions, such as drought stress [11,13,15,23,24] and dehydration kinetics [19]. In addition, THz-TDS technique can also investigate the structural behaviour and complex traits of leaves under any environment [14,19]. As compared to others, THz-TDS technique has proven to be more effective and reliable. However, the experimental setup of THz spectroscopy is not portable and requires a complex configuration of lasers [22].

In this paper, we present a novel, non-invasive approach to monitoring the WC of plant leaves using the scattering parameters of a THz pulse. Using a well-known material extraction algorithm, we computed permittivity from the scattering parameters for eight types of leaves, which we observed

for four consecutive days. The WC was then gauged from the decrease in the permittivity as the days passed. This paper is an expansion and presents a detailed analysis of our earlier work [8]. The rest of the paper is structured as follows: Section 2 describes the experimental setup followed by the material characterisation methods of plants leaves. Section 3 presents the measurement results and different parameters are discussed such as permittivity, the effect of weight and thickness, followed by a comparison of transmission response of all eight leaves between day 1 and 4. Finally, conclusion is discussed in Section 4.

## 2. Methods

### 2.1. Experimental Setup

We used a Swissto12 material characterisation kit (MCK) operating in the THz frequency range to obtain the scattering parameters of the plant leaves. The MCK was attached to a Keysight Technologies N5224A microwave network analyser (NA), the frequency range of which was shifted in the THz frequency range via a Virginia Diode vector NA extender module WM-250 (WR 1.0), enabling operation in the frequency range of 0.75 to 1.1 THz with a resolution of 2 GHz. The MCK comprised of two conical waveguide horn transitions with further two sections of the low-loss corrugated waveguide. A small aperture between the two low-loss corrugated waveguides allows the material samples to be inserted into the system during the measurement. Moreover, each half of the MCK comprises a waveguide which transitions from a rectangular waveguide at one end to a corrugated circular waveguide at the other. Furthermore, one half of the MCK, is fixed, while the other half is movable (to easily accommodate the insertion of the sample to be measured). To avoid any structural damage while the leaf specimen was clamped in the MCK for observation, we used two PTFE caps that enabled a uniform compression of the samples as shown in Figure 2. Prior to the measurement, the setup was configured using the two-port short-open-load-thru (SOLT) calibration technique.

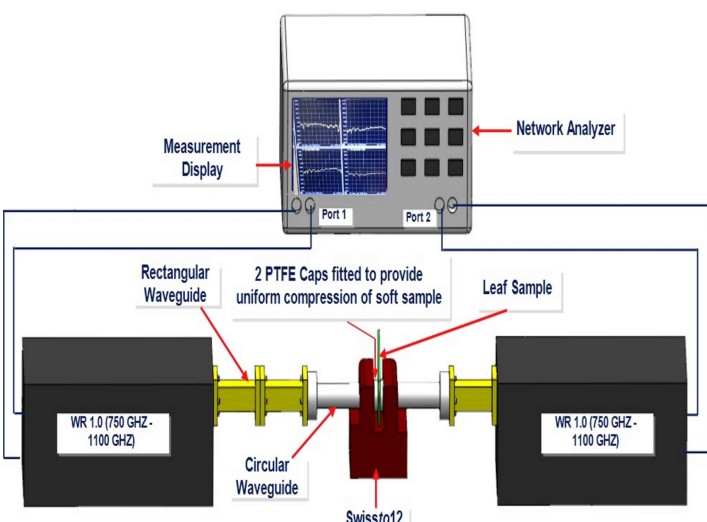

**Figure 2.** Schematic representation of experimental setup used for measurement of leaf sample. The leaf sample is placed between the two PTFE caps fitted to waveguide.

### 2.2. Sample Details

Eight different kinds of pot herbs were used, namely coffea arabica, aromatic coriander, basil, baby-leaf, pea-shoot, parsley, lamb's lettuce, and baby spinach. The fresh leaves were detached from the plants and placed in the laboratory for four consecutive days. The environment temperature for the measurements of leaves was $18.0\,^\circ\text{C} \pm 0.1\,^\circ\text{C}$, and the humidity was between $30\,\% \pm 2\,\%$. In this study, the weight and thickness of leaves were determined for four consecutive days using a precision electronic scale and Vernier caliper respectively. The leaves' thickness and weight were measured

every 2 h during the natural evaporation of leaf moisture. We used a Vernier scale to measure the leaf thickness and this process was repeated to determine thicknesses at three different locations to ensure the thickness was consistent across the surface of a leaf, which was in the range of 40 µm to 4 mm. The weight of leaf was measured using a digital kitchen scale with a least count of 0.1 mg. All leaves were measured at three different locations and on every location, four various orientations were considered to investigate the behaviour of leaves. From these observations, the purpose was mainly to determine any unevenness in the surface of leaves that may result in a change in the scattering response. For further illustration, Figure 3 shows the response of coffea leaf at three different locations which suggests that the orientation of leaf does not affect the measurements.

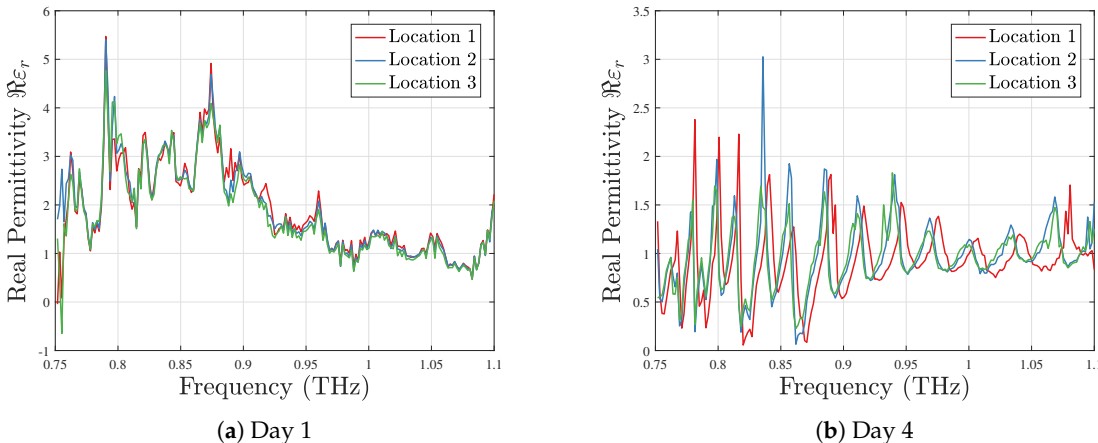

**(a)** Day 1          **(b)** Day 4

**Figure 3.** Real part of permittivity of coffea leaves at three different locations taken on days 1 and 4.

### 2.3. Material Characterization of Plant Leaves

The Nicholson-Ross-Weir (NRW) method [25] is the most common technique in which the dielectric parameters $\varepsilon_r$ and $\mu_r$ of a planar material are extracted from a two-port vector NA measurement in which the transmission and reflection coefficients are obtained through the S-parameters. This method belongs to the category of frequency-by-frequency material extraction in which every point from the frequency sweep is used. In general, the NRW method generates both the complex permittivity $\varepsilon_r = \varepsilon_r'' - j\varepsilon_r'$ and permeability, $\mu_r = \mu_r' - j\mu_r''$ of the specimen under test. Here, we assume that the leaves are non-magnetic and compute only the permittivity. One of the intrinsic problems of the NRW method is the periodicity of the phase of the electromagnetic wave that leads to ambiguous results. This problem has been discussed at length in other works [26–28]. To rectify this, we follow the step-wise approach in which the phase ambiguity is removed by using the phase delay information from the previous frequency point [29]. In this paper, we consider a plant leaf as a planar slab of thickness $d$ which is positioned between two air-filled circular waveguides. With the help of an equivalent transmission line model, the reflection ($\Gamma$) and transmission ($T$) coefficients of a semi-infinite slab are expressed in terms of the measured s-parameters, $S_{11}$ and $S_{21}$ as [30],

$$\Gamma = \chi \pm \sqrt{\chi^2 - 1}, \qquad\qquad T = \frac{S_{11} + S_{21} - \Gamma}{1 - (S_{11} + S_{21})\,\Gamma}, \qquad (1)$$

where the intermediate variable $\chi$ is defined as $\left(S_{11}^2 - S_{21}^2 + 1\right)/2S_{11}$. In the case of the slab with a finite thickness $d$, the transmission coefficient $T$ can be described in terms of the propagation constant, $\gamma$ as, $T = \exp(-\gamma d)$, which can subsequently be written in the Euler form as $|T|\exp(-j\phi)$ where $\phi$ denotes the phase term. The propagation constant is then determined using [28],

$$\gamma = \frac{1}{d}\{-\log(|T|) - j\phi + j2\pi n\} \text{ where } n \in \mathbb{Z} \qquad (2)$$

which results in an infinite number of branches of the complex valued root due to the logarithmic function, demonstrated by the presence of the $2\pi n$ term. The problem of selecting the proper branch is solved by the technique proposed in [29] in which at each frequency point, the phase delay information is recovered from the previous frequency point. If the phase difference, $\phi_i - \phi_{i-1} < \pi$, the method ensures the current branch is selected. The permittivity is then calculated by [31],

$$\varepsilon_r = \frac{\gamma}{\gamma_0}\left[\frac{1-\Gamma}{1+\Gamma}\right]. \tag{3}$$

## 3. Results and Discussion

In this paper, we aimed to determine the electromagnetic properties of leaves including permittivity, and physiological features such as weight and thickness that can affect the WC of leaves. In addition, a strong correlation between the determined properties and WC of leaves was observed. Furthermore, the transmission response of all eight leaves were investigated for four consecutive days.

### 3.1. Permittivity of Leaves

The complex-valued permittivity of eight different types leaves were extracted from measurements taken from three various locations with different percentage of WC in them. Furthermore, on every location, measurements were recorded using four different orientations of the leaves to observe any anisotropic behaviour. Figure 3a shows that for a coffea arabica leaf, neither the location, nor the leaf orientation had any effect on the real part of permittivity on day 1. However, the effect of location was notable on the day 4 as shown in Figure 3b. We believe that the drastic decrease in the leaf thickness is responsible for this behaviour.

Similar patterns were observed for other types of leaves as well. Figure 4 shows the real part of the permittivity for all the leaves measured on four consecutive days. It is significant to observe that all the leaves revealed the highest permittivity on day 1 when the WC in fresh leaves was the highest, and as the days progressed, permittivity showed a decrement when leaves became water stressed. Hence, dielectric parameter measurements differed significantly on days 1 and 4 for fresh, and water-stressed leaves. From these observations, it also showed a clear correlation between the permittivity and WC of leaves, i.e., fresh leaves with a higher amount of WC would have a high permittivity and vice versa. From Figure 5, it is evident that the real part of permittivity shows a strong decaying correlation with WC. As observed in Figure 4, various leaves also showed distinct decreasing responses from each other, attributed to different lead composition and structure. In another study, we have shown that the real part of permittivity can be used to classify leaves, that are described here with an accuracy of 98.2%.

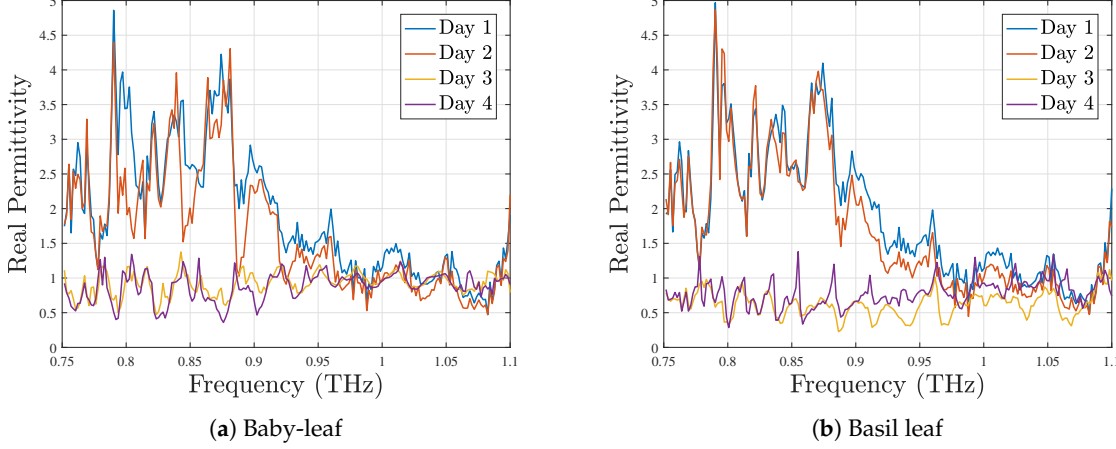

(a) Baby-leaf      (b) Basil leaf

**Figure 4.** *Cont.*

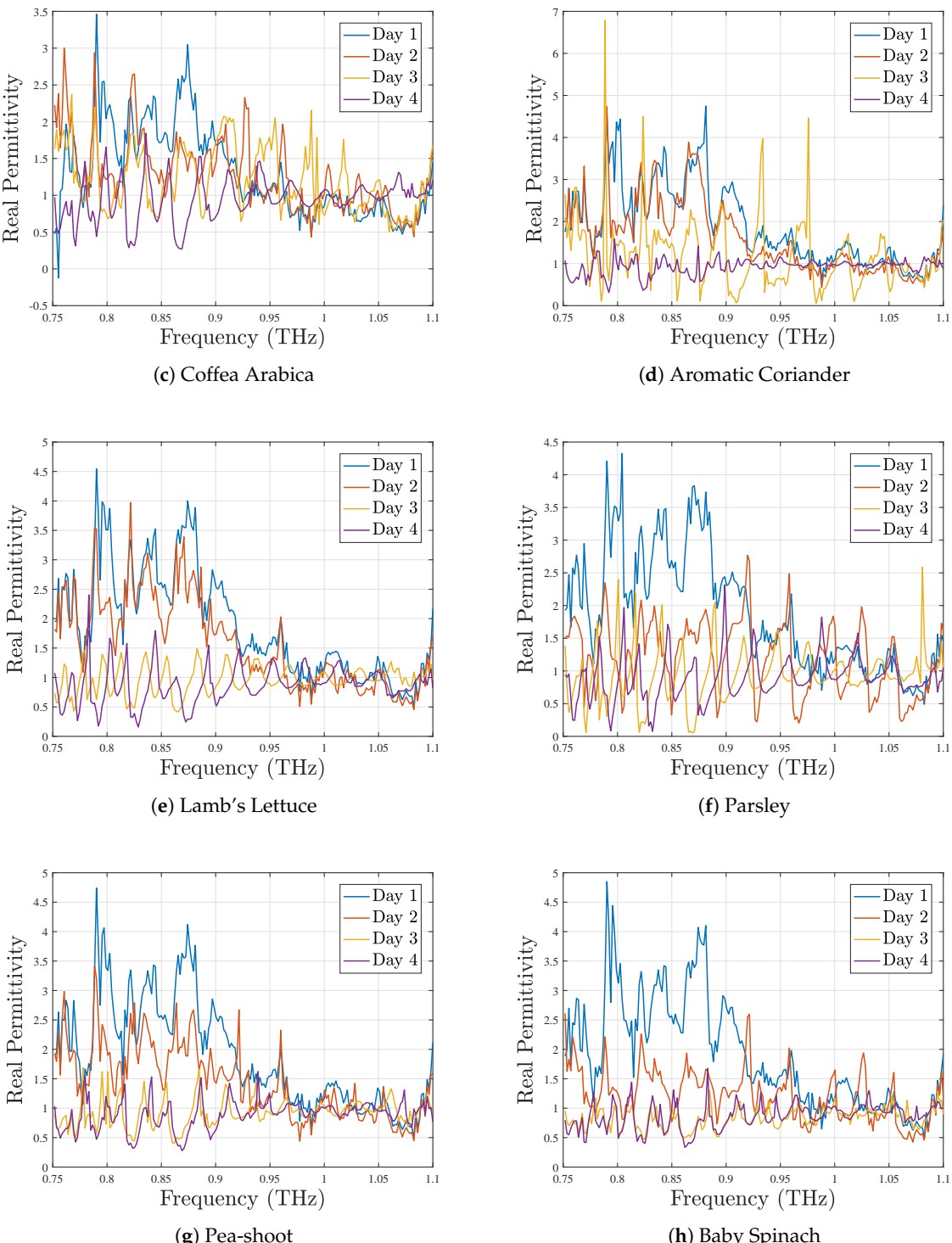

**Figure 4.** Real part of permittivity of all eight leaves measured on four consecutive days. Leaves become transparent to electromagnetic waves with the passage of days as seen by the decrease in the permittivity.

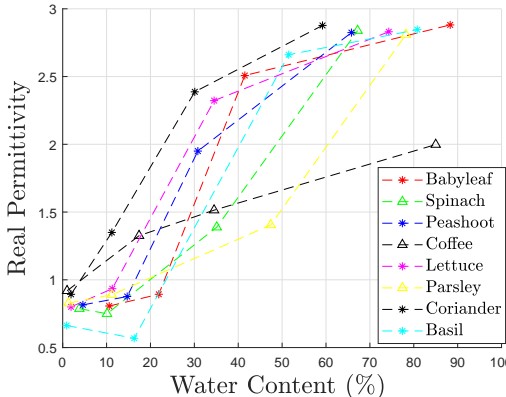

**Figure 5.** Correlation of permittivity with loss of WC in leaves.

### 3.2. Estimating Leaf Water Content

In this study, WC in leaves was observed by determining the physical parameters such as weight and thickness. Referring to the weight of leaves, initially on the first day, the time duration between the two weight measurements were maintained from two to three hours. On the second day, this was extended to four hours and finally, on the third and fourth day, it was increased to 6 hours. It was noted that there was significant decrease in the weights of some leaves as shown in Figure 6 on day 1, i.e., basil, baby-leaf and pea-shoot, whereas, other leaves displayed a slow decreasing trend in weight loss of leaves as days progressed.

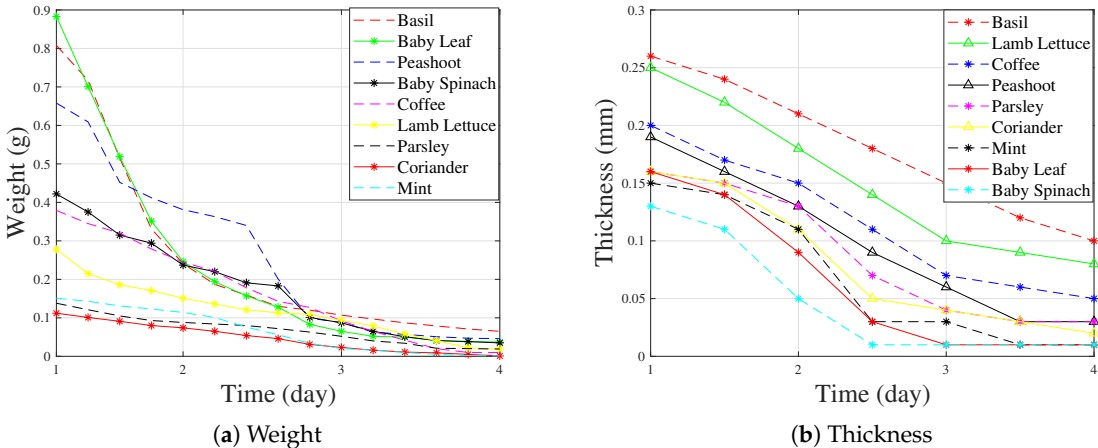

(**a**) Weight　　　　　　　　　　　　　　　　　　　(**b**) Thickness

**Figure 6.** Change in the physical properties of leaves with time.

This clearly indicated that the moisture in leaves evaporated more rapidly on the day 1 and 2 compared to day 3 and 4, thereby creating more air cavities in the leaves. To assess the variation of leaf WC during the leaf's water evaporation process, the measurements were translated into WC using [17,32],

$$WC = \frac{W_{\text{time}} - W_{\text{dry}}}{W_{\text{fresh}}} \times 100 \tag{4}$$

where $W_{\text{fresh}}$ is the weight of the fresh leaf, $W_{\text{time}}$ is the weight of a leaf measured over time and $W_{\text{dry}}$ is the weight of a dry leaf. In the beginning, the WC loss observed between the two hours on day 1 was found in the range of 5% to 22%. At the end of the investigation on day 4, this loss increased during the natural evaporation of leaf moisture and was established in the range of 83.12% to 99.33%. The obtained percentages loss of WC can be validated with Figure 6a. Considering this discussion, it also showed a significant correlation with the real part of the permittivity which was the highest on

the first day when the weight of leaves was considerably high compared with the fourth day as shown in Figure 4. The thickness of all the leaves was carefully determined to avoid any excess pressure to the samples that would cause disturbances in the morphological structure of the leaves, changing the dielectric properties of the samples as a result. As seen in Figure 6b, the thickness of leaves was considerably higher on day 1, implying a greater WC in fresh leaves compared to day 4 when mostly, all leaves were dried out. From this significant and meaningful observation, it was concluded that dehydration of leaves with passing days affected the thicknesses to a substantial degree. On day 4, some leaves stayed invariant or slight changes occurred in the thickness of leaves i.e., coriander and spinach as shown in Figure 6a. These transformations in the thickness of leaves evidently showed that WC in coriander and spinach leaves had evaporated to the maximum on day 4 and no further variations could be observed in thickness of leaves.

### 3.3. Evaluation of Leaf Transmission Response

In this section, transmission responses of all leaves were observed on day 1 and 4 as shown in Figure 7. It was noticed that on day 1, attenuations of all leaves were substantially high due to the presence of higher WC in tissues of leaves, which resulted in a higher absorption and lower transmission response. Moreover, on day 4, a substantial degree of increment in transmission response was observed as WC in leaves had evaporated to large extent, which resulted in a decrement of weight and thickness of leaves and eventually, less absorption occurred at this time. Figure 7 exhibited a strong correlation of transmission response with WC, weight and thickness of leaves. Baby-leaf exhibited a lower transmission response on day 1 compared to others, reflecting a higher presence of WC in leaf, which resulted in higher absorption. Contrarily, parsley displayed an increment in transmission response due to the presence of less WC in the leaf and hence, producing low absorption compared to other leaves.

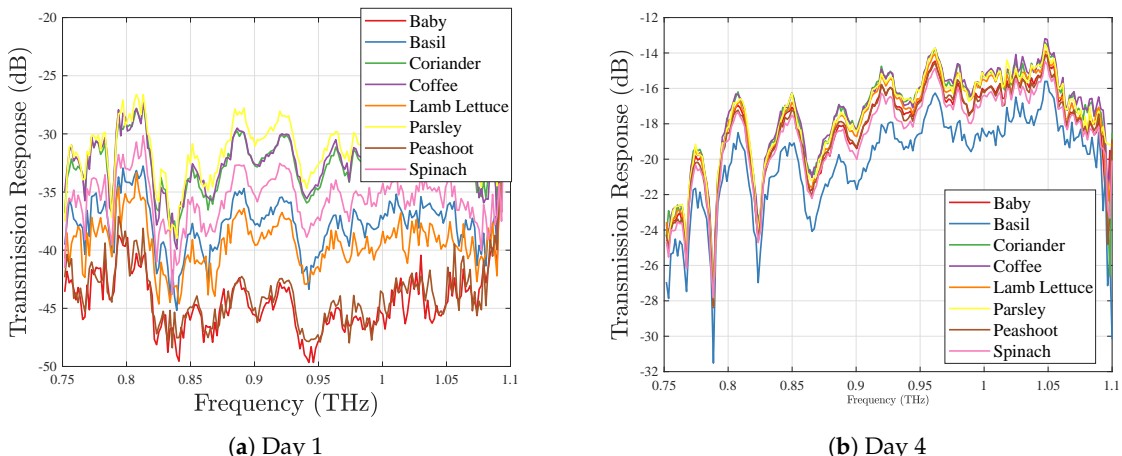

(**a**) Day 1  (**b**) Day 4

**Figure 7.** Transmission response of leaves on first and fourth days.

## 4. Conclusions

In this paper, a novel, non-invasive technique for characterising the water content (WC), and in turn the health of plant leaves was proposed using THz waves. The electromagnetic properties of eight types of leaves were determined for four consecutive days through the measured scattering parameters. The weight and thickness of the leaves were also recorded at the same time. We observed that the leaves became increasingly transparent to the terahertz (THz) waves through the course of four days experiment, as seen by the peaks in the real part of permittivity. Similar decaying trends were observed in the peak values of the real part of the extracted relative permittivity as the decreasing weight due to loss of WC. The significance of this paper lies in the simple, cost-effective technique and other advantages such as: (a) This paper proposes a unique technique to characterise and estimate WC

of eight various leaves in terms of electromagnetic parameters at THz frequency range from 0.75 to 1.1 THz. (b) The electromagnetic parameters are measured in simple, fast, and non-invasive manner using a THz material characterisation kit. Moreover, The structural integrity and configuration of leaves were also considered by employing two polytetrafluoroethylene (PTFE) caps which were fitted internally to the waveguide. (c) This paper establishes a notable correlation between electromagnetic parameters with WC in leaves i.e., change in the WC of leaves is reflected in the electromagnetic parameters at certain frequencies. In the age of a climate change driven water conservation, the proposed scheme can be used to design efficient irrigation systems on-site without any need to remove the leaves from plants.

**Author Contributions:** Conceptualization, A.Z., H.T.A. and Q.H.A.; software, A.Z. and H.T.A.; resources, D.R.S.C. and K.A.Q.; writing–original draft preparation, A.Z. and H.T.A.; writing–review and editing, Q.H.A., M.A.I. and A.A.; supervision, Q.H.A. and M.A.I.; project administration, Q.H.A.

**Funding:** This research was funded under EPSRC DTA studentship which is awarded to Adnan Zahid for his PhD.

**Conflicts of Interest:** The authors declare no conflict of interest.

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
