# Peer review of "Characterization and Water Content Estimation Method of Living Plant Leaves Using Terahertz Waves"

_applsci, doi:10.3390/app9142781_

Round 1
Reviewer 1 Report
Review of the manuscript untitled “Characterization and Water Content Estimation Method of Living Plant Leaves Using TerahertzWaves”
General comments
This study proposes to characterize and estimate water content of 8 various leaves in terms of electromagnetic parameters determined in the 0.75 to 1.1 THz frequency range by using a material characterization kit attached to a VNA. The author argues the novelty character of this approach and a notable correlation between electromagnetic parameters at THz frequencies and the water content of leaves. In order to be published in Applied Science, methods, results and conclusion must be improved in order to convince the readers of the correlation between THz e.m parameters and leaves WC and the pertinence of the THz approach compared to more conventional methods to track the moisture of plant leaves. In this goal, I proposed below several major and minor modifications for the submitted article.
Major modifications:
o Methods:
§ Several important information are missing in the experimental setup description. There is no mention about the absorption of atmospheric water vapour. Is the setup under vacuum? What kind of waveguides are used? What is their role? It is also important to precise the spectral resolution and the time averaging. What is the advantage of the VNA source with a rather limited spectral band compared to a THz-Time Domain Spectrometer which will provide a spectral band from 0.1 GHz up to several THz?
§ In the material characterization section, the permittivity calculation is detailed but the calculation of the correlation and of the refractive index (fig. 5 and 6) from the spectra are not explained.
o Measurements results:
§ In the sections 3.1 and 3.2, the authors must improve their discussion especially by comparing the THz spectra between the different leaves. The results presented in fig. 4, 5 and 6 show different time decays which are not some exemple of destructive methodscommented in the text.
§ The authors should explain the complementarity the THz e.m parameters (permittivity, refractive index and transmission response) for the WC monitoring.
§ In fig.7, it is surprising that the WC estimated with eq.4 is not plotted in addition to the weight and the thickness vs time. This WC obtained with the variation of the weight of the leaves must be compared with a WC directly determined from the THz e.m parameters in order to justify that the THz measurements are powerful to monitor WC in leaves. This is the main flaw of the publication which not provide a straightforward WC measurement from the THz data. The authors just argue a significant correlation between the real part of the THz permittivity and the WC estimated in leaves but this correlation is not demonstrated. A correlation diagram could be plotted to justify this statement.
§ In fig.8 absorption of water lines are clearly visible in the transmission response especially in the day 4 but there is no comment about these absorptions in the main text?
Minor modifications:
o In the introduction the authors have listed the application fields of THz waves but nothing is said about THz gas phase spectroscopy with numerous applications (astrophysics, pollution monitoring, breath analysis…). Moreover, the introduction can be improved by adding examples of destructive methods of WC monitoring and of elements highlighting advantages of the THz domain compared to other spectral domain measurement especially in the IR domain.
o From my opinion l. 51 to l. 58 providing the relevance of the article must be moved to the conclusion or to the cover letter but not in the introduction.
o The Fig. 2 shows the differences in permittivity depending on the location but not on the orientation as it is mentioned in the main text (p. 5 l.120).
o I don’t understand the usefulness of the figure 3. Moreover, the most of elements of this figure is not discussed in the text.
o I suggest to put on a unique page the figure 4 and to do the same with figure 6.
o The first sentence of section 3.3 and the sentence p.9 l.161-163 should be moved to section 2.2.

Reviewer 2 Report
The paper reports THz spectroscopy of plant leaves during the natural drying process. The authors observe a signature in the permittivity spectrum that vanishes after several days and assign it to the reduction of the water content in the leaves.
As it is outlined in the introduction, there were several previous studies using THz spectroscopy for the same purpose. However, the present approach relies on a microwave technique making the experimental setup rather compact and easily portable in contrast to femtosecond laser-based time-domain spectrometers. In this respect, the study represents an interest for the related scientific community.
I have got several questions and comments regarding the presented results:
1) There is a number of sharp features in the real and imaginary permittivity spectra and the authors do not try to assign them to any specific molecular vibrations in the leaves. Would it be possible or these are just some artefacts of the measurements? Has any test measurements on a well-defined reference substance like silicon or lactose been performed?
2) The water content is correlated only with the real part of the permittivity. However, the imaginary part also must show an enhanced absorption at low frequencies. I would recommend to show this data in the manuscript instead of the refractive index which basically reflects the same behavior as Re(epsilon).
3) Reference 18 reports THz imaging of leaves and not MRI and other techniques as stated in the manuscript.
In summary, the paper can be recommended for the publication after a proper revision.
Round 2
Reviewer 1 Report
Most of my recommandations have been take into account. I think that the paper could be now accepted.
Reviewer 2 Report
The manuscript can be published in the present form.